# Implementation of Medicalholodeck^®^ for Augmented Reality Surgical Navigation in Microsurgical Mandibular Reconstruction: Enhanced Vessel Identification

**DOI:** 10.3390/healthcare13192406

**Published:** 2025-09-24

**Authors:** Norman Alejandro Rendón Mejía, Hansel Gómez Arámbula, José Humberto Baeza Ramos, Yidam Villa Martínez, Francisco Hernández Ávila, Mónica Quiñonez Pérez, Carolina Caraveo Aguilar, Rogelio Mariñelarena Hernández, Claudio Reyes Montero, Claudio Ramírez Espinoza, Armando Isaac Reyes Carrillo

**Affiliations:** 1General Surgery Department, General Hospital of Chihuahua Dr. Salvador Zubirán Anchondo, Autonomous University of Chihuahua, Chihuahua 31200, Mexico; 2Oral and Maxillofacial Surgery Department, General Hospital of Chihuahua Dr. Salvador Zubirán Anchondo, Autonomous University of Chihuahua, Chihuahua 31200, Mexico; 3Plastic & Reconstructive Surgery Department, Central University Hospital Dr. Jesús Enrique Grajeda Herrera, Autonomous University of Chihuahua, Chihuahua 31200, Mexico

**Keywords:** augmented reality surgery, holographic navigation, microsurgical reconstruction, mandibular reconstruction, virtual surgical planning, free fibula flap

## Abstract

Mandibular reconstruction with the fibula free flap is the gold standard for large defects, with virtual surgical planning becoming integral to the process. The localization and dissection of critical vessels, such as the recipient vessels in the neck and the perforating vessels of the fibula flap, are demanding steps that directly impact surgical success. Augmented reality (AR) offers a solution by overlaying three-dimensional virtual models directly onto the surgeon’s view of the operative field. We report the first case in Latin America utilizing a low-cost, commercially available holographic navigation system for complex microsurgical mandibular reconstruction. A 26-year-old female presented with a large, destructive osteoblastoma of the left mandible, requiring wide resection and reconstruction. Preoperative surgical planning was conducted using DICOM data from the patient’s CT scans to generate 3D holographic models with the Medicalholodeck^®^ software. Intraoperatively, the primary surgeon used the AR system to superimpose the holographic models onto the patient. The system provided real-time, immersive guidance for identifying the facial artery, which was anatomically displaced by the tumor mass, as well as for localizing the peroneal artery perforators for donor flap harvest. A free fibula flap was harvested and transferred. During the early postoperative course and after 3-months of follow-up, the patient presented with an absence of any clinical complications. This case demonstrates the successful application and feasibility of using a low-cost, consumer-grade holographic navigation system.

## 1. Introduction

Mandibular defects resulting from tumor destruction create significant functional and aesthetic challenges that profoundly impact patient quality of life [1]. Osteoblastoma, an uncommon benign bone neoplasm accounting for approximately 1% of all primary bone tumors, exhibits a predilection for the spine and sacrum but can arise in virtually any bone [2]. Within the craniofacial region, it represents approximately 15% of cases, with the mandible being the most common site of involvement, particularly the posterior region [3]. The epithelioid variant of osteoblastoma presents unique diagnostic challenges due to its atypical epithelioid osteoblasts within a vascular stroma, which can mimic malignant proliferation [4]. Large tumors involving extensive portions of the mandible cause severe disfigurement of the lower facial third, with significant mandibular expansion potentially affecting the infratemporal fossa, parapharyngeal space, oral cavity, and floor of the mouth [5]. This cortical destruction leads to devastating functional consequences, including loss of bony continuity and structural support [6,7,8].

The fibula free flap has emerged as the gold standard for reconstructing composite bone and soft tissue defects in the head and neck region [9]. Since its initial description by Hidalgo in 1989, this microsurgical technique has become the optimal therapy for complex mandibular reconstructions worldwide [10]. The fibula’s favorable anatomical attributes include a straight, cortical bone segment with sufficient length (average 15.4 cm, up to 25 cm) to address extensive mandibular defects, excellent intrinsic rigidity from its compact bi-cortical structure, and dual blood supply that allows for multiple safe osteotomies [11,12]. These characteristics facilitate contouring the flap to mimic the mandibular shape and enable future implant placement [13]. Virtual surgical planning has become indispensable for these complex procedures, creating detailed digital replicas of patient anatomy that facilitate optimal surgical approach selection, treatment plan customization, precise definition of surgical procedures, assessment of desired resection margins, and simulation of surgery to anticipate potential anatomical or technical challenges [14].

Despite the sophistication of virtual surgical planning, a persistent challenge remains in transferring millimeter-level preoperative accuracy to the operating room environment. The successful execution of fibula free flap reconstruction critically depends on the accurate localization, dissection, and anastomosis of vessels at both donor and recipient sites [15,16]. Selection and preparation of recipient vessels in the head and neck region constitute a fundamental yet often demanding step, as the facial artery and internal jugular vein may be significantly compromised in patients requiring mandibular reconstruction. When ipsilateral vessels prove unsuitable due to their small caliber, fragility, radiation damage, or absence, surgeons must seek alternative recipient sites, potentially necessitating contralateral vessels, more proximal branches of the external or internal carotid system, or interposed venous grafts [17]. Similarly, intraoperative identification of cutaneous and muscular perforating vessels from the peroneal artery, which are vital for flap irrigation, presents significant technical challenges. The complex three-dimensional anatomy of this region complicates precise anatomical navigation during surgery, and conventional navigation systems often require visual diversion to external monitors, creating focus shift problems that may compromise surgical precision [18].

Augmented Reality (AR) represents a logical evolution beyond traditional navigation systems, offering the potential to superimpose digital information directly onto the surgeon’s view of the operative field, providing enhanced anatomical understanding and optimized precision [19]. Head-mounted displays promise direct integration of virtual anatomy data into the real-world surgical context, allowing more natural interaction without the need for external monitor consultation [20,21,22,23]. This technology enables a form of enhanced visualization that can facilitate intraoperative decision-making and ultimately improve clinical outcomes [24]. The objective of this report is to describe the first surgical intervention in Latin America using a holographic surgical navigation system for intraoperative planning and real-time vascular identification during mandibular reconstruction using a fibula free flap, representing a significant advancement in translating virtual surgical planning accuracy to the operating room environment. Informed consent was taken from the patient themselves. The following case report was elaborated on in alignment with the CARE guidelines and checklist [25].

## 2. Case Presentation

A 26-year-old previously healthy female presented with a three-month history of progressive left submandibular swelling. Initial symptoms included localized temperature changes and mild masticatory discomfort. The mass demonstrated progressive enlargement with associated pain and significant aesthetic concern prompting medical evaluation. Physical examination by the oral and maxillofacial surgery service revealed a firm, immobile 5 cm × 3 cm × 5 cm submandibular mass adherent to deep tissue planes with tenderness on palpation. No cervical lymphadenopathy or facial nerve dysfunction was detected. Contrast-enhanced computed tomography demonstrated a 6 cm × 3 cm × 5 cm osteolytic lesion of the left mandibular body with characteristics consistent with high tumor vascularity and associated cervical lymph node reaction. Importantly, the lesion demonstrated an intimate anatomical relationship with the facial and mandibular arteries without evidence of direct vascular invasion or compromise.

### 2.1. Holographic Navigation Workflow

Contrast-enhanced CT angiography of the lower extremities confirmed adequate bilateral vascular patency without atherosclerotic plaques or compromised blood flow. The study documented at least three muscular perforating arteries in each lower extremity with optimal visualization of the peroneal artery pedicle without evidence of anatomical variants or vessel foreshortening. The patient was admitted 24 h prior to surgery for innovative virtual surgical planning using AR technology. Meta Quest 3 headsets (Menlo Park, CA, USA) were utilized in conjunction with Medicalholodeck^®^ (Technoparkstr, Zurich, Switzerland) (Medical Imaging software version 1.5.0) to generate three-dimensional holographic models through a systematic five-step workflow process. The initial DICOM import phase involved direct integration of previously acquired CT datasets into the Medicalholodeck^®^ platform, maintaining the original image resolution and spatial coordinates. The segmentation process employed automated algorithms with manual refinement to isolate anatomical structures of interest, including vascular networks, bone architecture, and soft tissue boundaries. Advanced volumetric rendering techniques converted the segmented data into interactive three-dimensional models during the model generation phase, utilizing the machine learning-based tissue classification algorithm TotalSegmentator to differentiate between anatomical components [26]. The planning phase enabled virtual marker placement, measurement tools, and surgical approach simulation within the augmented reality environment. Following completion of virtual planning, the export phase generated standardized file formats compatible with intraoperative systems, while intraoperative registration allowed real-time alignment of holographic models with patient anatomy. Registration methodology incorporated fiducial marker-based calibration using anatomical landmarks identified during preoperative imaging. The system achieved a target registration error of less than 2.5 mm between virtual models and Doppler ultrasound marked by another surgeon. Calibration procedures were performed immediately prior to surgical intervention using standardized reference points on the patient’s craniofacial skeleton and lower extremity bony landmarks. The surgical workflow commenced with sterile draping protocols that maintained visualization access to critical anatomical regions while preserving aseptic conditions. The attending surgeon served as the primary system operator wearing the Meta Quest 3 headset, with dedicated technical support personnel managing headset calibration and software navigation. A secondary surgeon participated in the procedure through real-time broadcasting capabilities, providing continuous feedback and guidance to the primary surgeon. The entire surgical procedure was recorded through the Meta Quest 3 headset’s capture functionality, documenting both the augmented reality overlay and the actual surgical field for subsequent analysis and educational purposes (Figure 1). Workflow verification involved systematic confirmation of anatomical landmark alignment, measurement validation against physical examination findings, and cross-referencing with preoperative imaging studies. The collaborative surgical approach enabled immediate consultation and decision-making support, with the broadcasting surgeon providing real-time anatomical guidance and technical recommendations throughout the procedure. Independent holograms of the craniofacial region and lower extremities were generated, enabling detailed anatomical visualization of critical structures. The craniofacial hologram precisely identified the facial artery origin in the distal two-thirds of the left mandibular ramus, demonstrating lateral deviation secondary to mass effect from the tumor. Virtual markers were established through the software’s annotation tools to delineate the exact vascular origin site and guide optimal surgical incision placement. For donor site planning, the lower extremity hologram enabled precise identification of the bilateral perforating arteries, prioritizing the extremity contralateral to the lesion side. Holographic model superimposition over the patient’s anatomy facilitated cutaneous marking of the peroneal artery pedicle and muscular perforators using sterile surgical markers.

### 2.2. Surgical Technique

The patient was positioned supine under general anesthesia with nasotracheal intubation. Two simultaneous surgical teams were established: the tumor resection and mandibular reconstruction team, and the free fibular flap harvest team. During the resection phase, the primary surgeon utilized AR headsets to superimpose the hologram over the patient’s anatomy. This technology enabled precise tumor resection margin marking and real-time identification of critical vascular landmarks during dissection. A 6 cm left submandibular incision was performed followed by layer-by-layer dissection using monopolar and bipolar electrocautery. Throughout the dissection, the AR system provided continuous notifications regarding facial artery trajectory. Upon reaching the deep plane, the facial artery and accompanying veins were identified and isolated, preserving these structures during osteotomy and tumor resection. The resulting mandibular defect was measured using the millimetric ruler incorporated within the AR software, documenting an approximately 12 cm long defect. The primary surgeon, utilizing AR headsets, transitioned to the flap harvest team to delineate the exact dimensions of the required bone segment. Communication to the team specified the need for a 12 cm fibular segment with three preplanned osteotomies marked both in the application and on the flap. A 12 cm osteo-fascial flap was successfully harvested without the skin component, preserving the arterial and venous pedicle. Subsequently, graft modeling was performed to recreate mandibular ramus and angle anatomy, achieving an L-shaped configuration. Bone graft fixation was accomplished using titanium plating on the external cortex. Flap revascularization was achieved through end-to-end microvascular anastomosis using interrupted 8-0 nylon suture technique for both arterial and venous facial vessels. Closure was performed in anatomical layers: oral mucosa with 4-0 catgut, subcutaneous tissue with absorbable suture, and skin with continuous 3-0 polypropylene intradermal suture. At the lower extremity donor site, primary wound closure was performed with split-thickness skin graft placement for distal defect closure measuring approximately 5 cm.

The histological examination revealed the presence of a cellular infiltrate whose initial morphological characteristics suggested a possible histiocytic process (Figure 2). However, the complexity of the observed cellular pattern required complementary studies using immunohistochemical techniques to establish an appropriate differential diagnosis and to determine the precise nature of the lesion. The immunohistochemical studies provided definitive diagnostic information through the evaluation of multiple specific markers. The assessment of CD68 was entirely negative, thereby conclusively ruling out any histiocytic-type differentiation. Similarly, the S-100 protein showed absolute negativity, thus excluding a neural or melanocytic origin for the lesion. Conversely, the analysis of SATB2 demonstrated intense and diffuse expression with a maximum score, unequivocally confirming the osteoblastic differentiation of the tumor cells.

### 2.3. Postoperative Course

A nasogastric tube was placed immediately postoperatively for enteral access maintenance. During the first 24 h, the patient experienced a fever of 37.8 °C, which resolved with antipyretic therapy without additional clinical manifestations. On postoperative day three, a head and neck CT was acquired to assess the viability of the microvascular flap and microvascular anastomosis (Figure 3).

Oral feeding was initiated with liquids at 48 h postoperatively, progressing satisfactorily to juices, teas, and water, permitting nasogastric tube removal. On postoperative day three, pureed foods were introduced and assisted ambulation was initiated by the physical medicine and rehabilitation service. By postoperative day five, the patient advanced to soft diet consistency without evidence of food intolerance, masticatory pain, or dysphagia. Hospital discharge occurred on postoperative day seven without complications. At three months postoperatively, the patient returned for ambulatory follow-up demonstrating near-complete donor site wound healing in the left leg and complete submandibular wound healing (Figure 4). No purulent drainage, wound dehiscence, or infection signs were evident. The patient continued physical rehabilitation to resume complete preoperative physical activity, reporting a return to occupational and personal activities independently without additional assistance requirements. Ongoing consultations with the oral and maxillofacial surgery service have been established for follow-up care and management.

## 3. Discussion

This case represents the first successful application of low-cost holographic surgical navigation technology for complex microsurgical mandibular reconstruction in Latin America, demonstrating the feasibility of translating advanced augmented reality capabilities into routine clinical practice. The study by Pereira et al. provides another example of augmented reality applications for flap surgery in Latin America. However, their research developed a different approach for microsurgical planning through smartphone-based augmented reality. Their system superimposes computed tomography angiography (CTA) reconstruction images of vascular anatomy directly onto the smartphone camera interface, enabling guided delineation of perforating vessels and lymph nodes to serve as a precise, portable “dissection roadmap” for surgeons [27]. While innovative, this study utilized only smartphone technology without the immersive three-dimensional visualization and pass-through capabilities available in more advanced systems. In contrast, the Meta Quest 3 headset combined with Medicalholodeck^®^ software provided real-time anatomical guidance that enhanced surgical precision during both tumor resection and fibula free flap harvest, establishing a novel paradigm for intraoperative navigation in reconstructive microsurgery. The three-dimensional holographic visualization provided unprecedented anatomical detail, enabling precise identification of critical vascular structures and optimal surgical approach planning [28]. This represents a significant advancement in the integration of augmented reality technology within complex surgical procedures.

Real-time intraoperative guidance through AR technology facilitated accurate tumor resection margins and preserved vital anatomical structures, particularly the facial artery which showed lateral deviation due to tumor mass effects. The observed advantages of holographic navigation extend beyond traditional surgical planning methodologies. The technology enabled precise real-time identification of critical vascular structures, particularly the facial artery trajectory, which was altered by tumor mass effects, and would have required extensive dissection using conventional techniques. Unlike handheld Doppler ultrasonography, which provides auditory feedback requiring interpretation and offers limited spatial resolution, the holographic system delivered a direct visual overlay of vascular anatomy onto the operative field [29]. This enhanced anatomical precision, potentially reduced vascular dissection time, and increased surgeon confidence during critical phases of the procedure [30]. Furthermore, the technology served as an invaluable educational tool for surgical residents, providing three-dimensional anatomical visualization that enhanced understanding of complex microvascular relationships without compromising sterile technique or requiring additional personnel for navigation system operation.

AR technology can be compared with color Doppler ultrasound (CDU) in flap surgery regarding its precision in perforator identification, as well as the number of perforators identified, surgical time, and complications. Research studies suggest that AR, particularly when combined with artificial intelligence algorithms, offers significant advantages, although limitations remain. Studies indicate that AR, utilizing computed tomography angiography (CTA) data and algorithms, demonstrates superior precision in localizing perforating vessels compared to color Doppler [31]. One study reported a recognition rate of 94.3% for the AR group (AR with algorithms) versus 82.0% for the CDU group (*p* = 0.008). The average distance between the marked points and actual exit points was 1.5 mm for the AR group compared to 2.7 mm for the CDU group (*p* < 0.0001), representing a 1.2 mm reduction in localization error [32]. Another study suggests that AR provides precision benefits of nearly 2 mm compared to conventional Doppler techniques, which is significant given that the perforator size typically ranges from 1 to 2 mm. AR helps prevent erroneous judgments and “blind” exploration of perforating vessels, thereby minimizing unnecessary subcutaneous tissue dissection [33]. While CDU provides detailed hemodynamic information and can detect vessels approximately 0.5 mm in diameter, it may generate high rates of false positives and false negatives. Its accuracy depends heavily on operator skill and does not reflect the anatomical details of source arteries [34]. Some researchers suggest that AR and virtual reality (VR) offer limited benefit in terms of precision compared to CTA (which has an average precision of 2.3–5 mm) and color Doppler (6 mm). The lack of consistency in reported results and the heterogeneity of technologies utilized limit the reliability of a definitive evaluation, requiring higher-quality studies [35]. No significant difference has been found in postoperative complication rates between AR and color Doppler. A meta-analysis found that AR use did not significantly affect the risk of complications such as wound dehiscence, revision or flap loss, or infection (OR 0.6; 95% CI 0.3–1.3; *p* = 0.20) [33].

AR is associated with a greater number of intraoperatively identified perforators compared to conventional imaging methods. In one study, the magnetic resonance imaging (MRI) group identified 100 out of 106 actual vessels (a detection rate of 94.3%), whereas the color Doppler ultrasound (CDU) group identified 106 out of 128 actual vessels (a detection rate of 82.0%). The detection rate for the MRI group was significantly higher (*p* = 0.008) [36]. Identifying more perforators provides surgeons with more options for flap design, including the size and extension of the skin paddle, the length of the pedicle, and the angle of dissection, which may reduce overall morbidity. AR has been shown to significantly decrease both the flap harvest time and total operative time [37]. The mean flap harvest times were 52 min for the MRI group versus 68 min for the CDU group (*p* < 0.0001). A reduction of 19 min (12%) in flap harvest time was observed in a randomized controlled trial that utilized projection-based AR versus preoperative ultrasound mapping. Improved efficiency in flap reconstruction is associated with a lower rate of complications and a shorter hospital stay. The benefit in perforator identification time must be weighed against the time required to set up the AR environment. The learning curve for the technology can be up to 30 min, in addition to a processing time of approximately 40 min, which adds a significant temporal consideration [29].

Several technical limitations persist in current AR implementations for surgical procedures. Time delays remain a significant concern, along with inferior image quality compared to contemporary monitors. The weight of portable devices presents ergonomic challenges for extended surgical procedures, while the need for more intuitive and specialized software continues to limit widespread adoption. Additionally, AR holograms encounter difficulty adapting precisely to tissue deformation and depth variations during surgery. Environmental factors within the operating room create additional obstacles. Ambient lighting can interfere with AR projection, requiring lights to be dimmed or turned off, which may disrupt surgical workflow [34]. The precise alignment of perforating vessels with their three-dimensional models represents a key challenge. Manual registration methods lack automation and remain subject to subjective errors, while adhesive positioning devices may be affected by the patient’s tissue displacement [29]. AR implementation involves substantial initial hardware acquisition costs, with devices such as the Microsoft HoloLens (Microsoft Corporation, Redmond, WA, USA)costing up to USD 4000 per unit, in addition to software licensing expenses. Custom software development also represents a significant investment, requiring approximately USD 16,500 USD and two months of development time for a single platform [32]. However, some studies have reported economic benefits through cost reductions. These include savings of EUR 298 euros per flap with AR implementation and EUR 1000 per patient with virtual reality applications, primarily attributed to reduced surgical time. These potential savings suggest that despite high initial costs, AR technology may provide long-term economic advantages through improved surgical efficiency and reduced operative durations [34]. As a single case report, these findings cannot be generalized to broader patient populations or diverse anatomical presentations. The learning curve associated with holographic navigation technology presents challenges for both primary surgeons and surgical team members, requiring dedicated training time and technical proficiency development. Additionally, prolonged use of head-mounted displays during extended microsurgical procedures may cause ergonomic discomfort or visual fatigue that could potentially impact surgical performance, though this was not observed in the current case. The integration of AR technology in surgical practice represents an evolving field with expanding applications across multiple specialties [28].

While limited reports exist regarding holographic navigation in head and neck reconstruction, similar technologies have demonstrated promise in neurosurgery, orthopedic surgery, and cardiac procedures. The accessibility and affordability of consumer-grade AR devices, such as the Meta Quest 3, democratize advanced surgical navigation capabilities that were previously limited to specialized centers with expensive dedicated systems. Future applications of this technology may include complex pelvic reconstruction, breast reconstruction with perforator flaps, and other microsurgical procedures requiring precise anatomical navigation. Validation of this technique through multicenter studies with larger patient cohorts will be essential to establish standardized protocols, determine optimal indications, and quantify improvements in surgical outcomes, operative time, and complication rates.

## 4. Conclusions

This case demonstrates that holographic intraoperative navigation using commercially available devices represents a paradigm shift in surgical planning technology with significant potential to enhance outcomes in reconstructive microsurgery. The successful integration of consumer-grade augmented reality into complex mandibular reconstruction establishes a foundation for broader implementation across surgical specialties. While limitations exist, the demonstrated feasibility, precision, and accessibility of this approach warrant continued investigation through rigorous clinical trials to establish comprehensive protocols and validate long-term outcomes in diverse patient populations. The utilization of Meta Quest 3 headsets coupled with Medicalholodeck^®^ software demonstrated the feasibility of translating sophisticated preoperative planning into precise intraoperative execution, addressing the critical gap between virtual surgical planning and real-time operative guidance.

The integration of artificial intelligence algorithms, including TotalSegmentator for automated tissue classification, represents a significant advancement in surgical planning accuracy. The systematic five-step workflow process, from DICOM import through intraoperative registration, provides a standardized framework for implementing holographic navigation across diverse surgical specialties. The broadcasting capability enabling real-time consultation represents an additional innovation with implications for surgical education and remote expert guidance. Future research directions should focus on multicenter validation studies with standardized outcome metrics, including operative times, complication rates, and functional outcomes. The development of automated registration algorithms could address current limitations in manual calibration procedures. Integration with real-time imaging modalities such as intraoperative fluorescence angiography could further enhance the precision of vessel identification and anastomotic assessment. The convergence of advanced imaging, artificial intelligence, and immersive visualization technologies positions holographic navigation as a transformative tool in the evolution of precision surgery.

## Figures and Tables

**Figure 1 healthcare-13-02406-f001:**
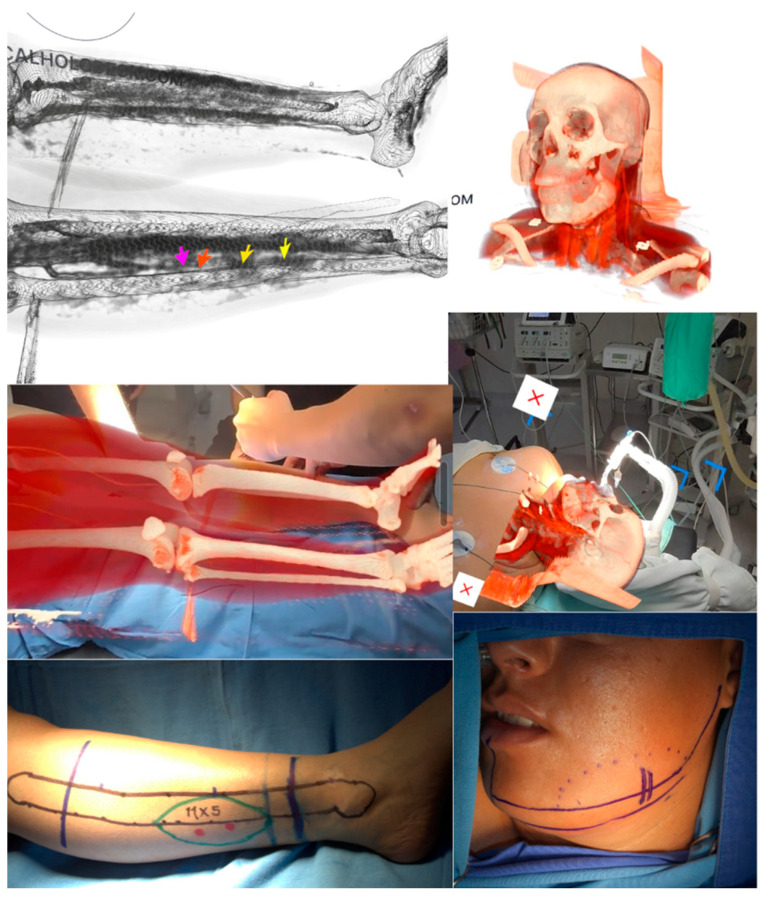
Surgical planning and microsurgical execution using 3D holographic navigation for reconstruction of mandible with free fibula flap. Intraoperative projection of a 3D hologram derived from CT reconstruction with the Medicalholodeck^®^ software. The arrows demonstrate the course and branching patterns of perforating vessels. Imaging quality and perforator visibility are significantly enhanced when contrast timing is optimized for arterial phase imaging, which is typically obtained 15–20 s after contrast injection. Holographic image of the skull and lower extremities is superimposed in real-time over the patient in the operating room, enabling advanced anatomical visualization and precise navigation. Preoperative surgical marking guided by 3D reconstruction. Anatomical site of origin of the facial artery is precisely delineated, and cutaneous perforating arteries of the peroneal artery in the lower extremity are identified and marked.

**Figure 2 healthcare-13-02406-f002:**
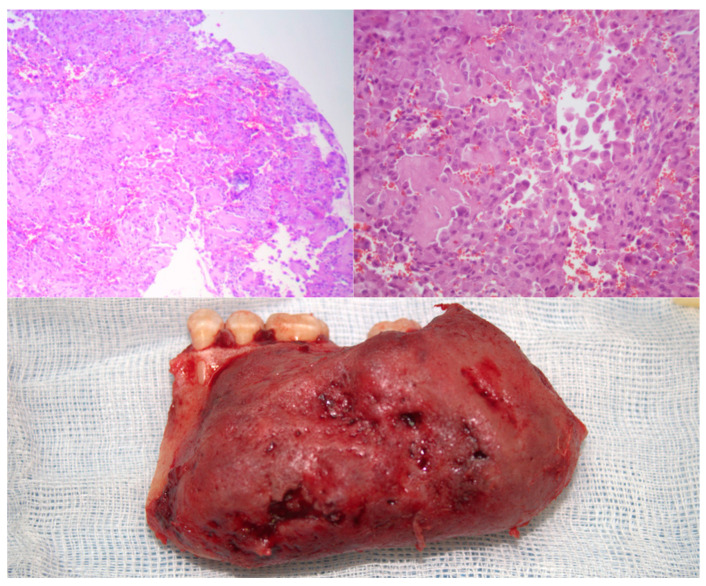
Histopathological and macroscopic correlation of mandibular osteoblastoma. Photomicrographs of the tumor lesion with hematoxylin and eosin (H & E) staining at 40× magnification. Presence of immature bone and osteoid trabeculae surrounded by a vascularized stroma rich in epithelioid-appearing osteoblasts. Macroscopic image of the surgical specimen presenting an exophytic tumor mass is shown, with inflammatory appearance.

**Figure 3 healthcare-13-02406-f003:**
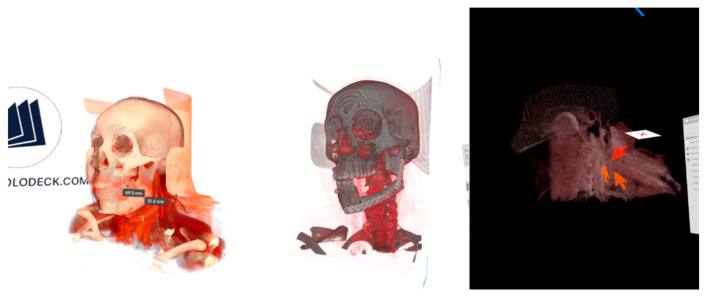
Preoperative and postoperative control of free fibula flap reconstruction. Three-dimensional holographic model demonstrating the postoperative result. Arrows indicate adequate blood flow at the junction of the flap’s peroneal artery with the facial artery. Holographic 3D models made with Medicalholodeck^®^ rendering software. The arrows indicate the microvascular anastomotic site, which demonstrate patent vessel continuity without evidence of stenosis or thrombotic complications. The imaging findings confirm adequate perfusion throughout the reconstructed tissue, suggesting successful microvascular anastomosis and viable flap integration at the recipient site.

**Figure 4 healthcare-13-02406-f004:**
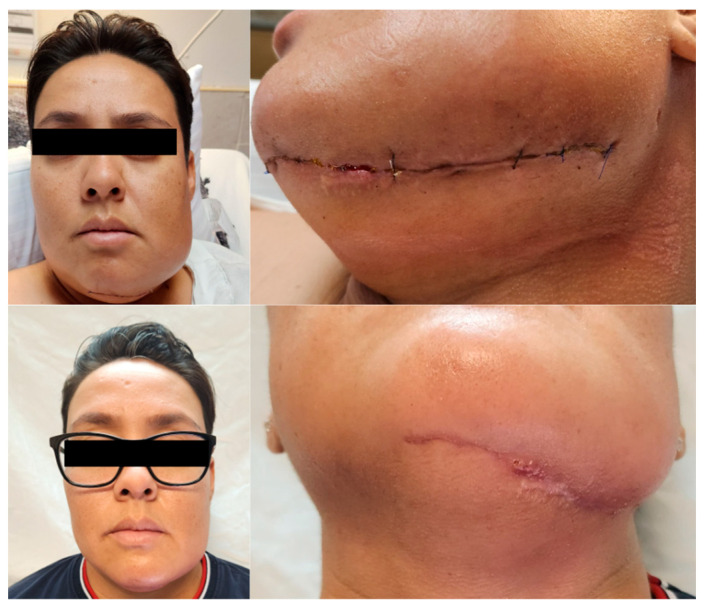
Clinical follow-up of postoperative mandibular resection. Evolution of facial edema. A comparison is made between the marked facial inflammation at 24 h postoperatively and the notable improvement with absence of significant inflammation at the 3-month follow-up.

## Data Availability

Data are contained within the article.

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
