# Peer review of "Implementation of Medicalholodeck^®^ for Augmented Reality Surgical Navigation in Microsurgical Mandibular Reconstruction: Enhanced Vessel Identification"

_healthcare, 2025, doi:10.3390/healthcare13192406_

Round 1
Reviewer 1 Report
Comments and Suggestions for Authors
It should follow CARE guidelines.

Author Response
Thank very much, for taking you valuable time as a reviewer, it means a so much to us a researcher to get feedback and improve our academic writting.
Comment 1: The manuscript is a case report. We don't find those points to ponder. It should follow the CARE guidelines; that is enough to say the points that the authors need to look into in the manuscript. Then only the thorough review can be done. Hence the review is very short.
Response: We aknowledge your comment, and the authors, revised the manuscript to be correctly aligned to the CARE guidelines, to address, we downloaded the CARE guideline template, to check every box, and verify its correctly done our case report.

Reviewer 2 Report
Comments and Suggestions for Authors
This is an interesting and well-written case report describing the first reported use of a consumer-grade holographic navigation system in Latin America for microsurgical mandibular reconstruction. The manuscript is innovative, clinically relevant, and provides a detailed step-by-step description of planning, surgical execution, and short-term outcomes.
There are, however, some points that could improve the clarity and balance of the paper:
-
Abstract – Please consider adding brief information about the postoperative course and follow-up, and explicitly acknowledge the limitation of being a single case report.
-
Figures – The captions are very detailed and sometimes repetitive. More concise captions would improve readability. Also, please clarify whether intraoperative images are original photographs or software renderings.
-
Discussion – While comprehensive, the discussion could be more balanced. A specific paragraph addressing limitations (e.g., registration accuracy, learning curve, ergonomic issues, comparison with standard techniques such as Doppler or conventional navigation) would strengthen the manuscript.
-
Follow-up – Only the three-month follow-up is described. If longer follow-up data are available, even briefly, please include them. If not, a statement on ongoing follow-up would be appropriate.
-
Language and style – The manuscript is clear, but some sentences are long and redundant. Minor language editing would improve fluency.
Overall, this is a valuable and original contribution. With minor revisions to highlight limitations, streamline figures, and refine the discussion, the manuscript will be suitable for publication.
Comments on the Quality of English LanguageThe English is generally clear and understandable. However, several sentences are long and could be simplified to improve readability. Minor editing is recommended to reduce redundancy and enhance fluency, particularly in the Introduction and Discussion sections.
Author Response
Thank you very much, taking time to review our manuscript. We addressed every comment from the review documento, and we mention point by point every changed made in the manuscript.
Comment 1: Please consider adding brief information about the postoperative course and follow-up, and explicitly acknowledge the limitation of being a single case report.
Response: We added in the revised manuscript the following text to address this comment: During early postoperative period and after 4-month follow-up the patient presented with absence of any clinical complication.
Comment 2: The captions are very detailed and sometimes repetitive. More concise captions would improve readability. Also, please clarify whether intraoperative images are original photographs or software renderings.
Response: We revised very carefully every caption of the figures, and we made corrections to restrict the number of words and lenght of every caption.
Comment 3: While comprehensive, the discussion could be more balanced. A specific paragraph addressing limitations (e.g., registration accuracy, learning curve, ergonomic issues, comparison with standard techniques such as Doppler or conventional navigation) would strengthen the manuscript.
Response: We addressed your comment by realized an additonal research about the the comparision of standard techniques for identifying perforators with doppler and virtual reality. We made a whole paragraph about this subject to improve our manuscript.
Comment 4: Only the three-month follow-up is described. If longer follow-up data are available, even briefly, please include them. If not, a statement on ongoing follow-up would be appropriate.
Response: We added in the text of the manuscript the statement that on goin follow-up is on course in the Oral and Maxillofacial consultation.
Comment 5: The manuscript is clear, but some sentences are long and redundant. Minor language editing would improve fluency.
Response: We as the authors, revised the manuscript with an english editing service to improve fluency and have better impact for possible readers.
Reviewer 3 Report
Comments and Suggestions for Authors
Dear respected authors. By addressing the following points, the clarity and impact of your manuscript can be significantly improved, facilitating better understanding and engagement from readers and reviewers.
- A self-contained, descriptive title with a clearly indicated study type can greatly improve your paper's clarity and reception. It does not clearly state the main objective or outcome being evaluated.
- In methodology section: AR (Augmented Reality) in scientific or clinical writing, it's essential to spell them out on first use to ensure clarity for all readers.
The study involves live human participants informed consent must be obtained, and this should be explicitly stated. If it does not, you should clarify why it was not needed
3.In discussion section; ‘This case represents the first successful application of low-cost holographic surgical navigation technology for complex microsurgical mandibular reconstruction in Latin America, demonstrating the feasibility of translating advanced augmented reality capabilities into routine clinical practice.’
The sentence as it stands is ambiguous about whether the technology is novel to the investigators only, or if it is genuinely the first time anyone has used it for this indication in Latin America (or more broadly)
With claims of global novelty — they require thorough literature review and strong evidence.
4.In conclusion section: You stated that; Holographic intraoperative navigation using commercially available devices represents an exciting and accessible evolution of surgical planning technology…’’ This refers to the instruments or platforms used in the study (e.g., Microsoft HoloLens, AR software, etc.).This detail supports the accessibility and practicality of the technique but does not define the goal of the study.
Author Response
Response to reviewer
Comment 1: A self-contained, descriptive title with a clearly indicated study type can greatly improve your paper's clarity and reception. It does not clearly state the main objective or outcome being evaluated.
Response: The title of the manuscript was modified by the authors to improve and clarify the whole purpose of the manuscript.
New title: Implementation of Medicalholodeck® for Augmented Reality Surgical Navigation in Microsurgical Mandibular Reconstruction: Enhanced Vessel Identification
Comment 2: In methodology section: AR (Augmented Reality) in scientific or clinical writing, it's essential to spell them out on first use to ensure clarity for all readers.
Response: This spelling problem was addressed and corrected in the text, and was added to the revised version of the manuscript.
Comment 3: The study involves live human participants informed consent must be obtained, and this should be explicitly stated. If it does not, you should clarify why it was not needed
Response: An acknowledgment about the participant informed consent was added to the manuscript in the introduction section.
Comment 4: In discussion section; ‘This case represents the first successful application of low-cost holographic surgical navigation technology for complex microsurgical mandibular reconstruction in Latin America, demonstrating the feasibility of translating advanced augmented reality capabilities into routine clinical practice.’
Response: This case represents the first successful application of low-cost holographic surgical navigation technology for complex microsurgical mandibular reconstruction in Latin America, demonstrating the feasibility of translating advanced augmented reality ca-pabilities into routine clinical practice. The study by Pereira et al. provides another example of augmented reality application for flap surgery in Latin America. However, their research developed a different approach for microsurgical planning through smartphone-based augmented reality. Their system superimposes computed tomog-raphy angiography (CTA) reconstruction images of vascular anatomy directly onto the smartphone camera interface, enabling guided delineation of perforating vessels and lymph nodes to serve as a precise, portable "dissection roadmap" for surgeons [36]. While innovative, this study utilized only smartphone technology without the immer-sive three-dimensional visualization and pass-through capabilities available in more advanced systems. In contrast, the Meta Quest 3 headset combined with Medicalho-lodeck® software provided real-time anatomical guidance that enhanced surgical pre-cision during both tumor resection and fibula free flap harvest, establishing a novel paradigm for intraoperative navigation in reconstructive microsurgery.
Comment 5: In conclusion section: You stated that; Holographic intraoperative navigation using commercially available devices represents an exciting and accessible evolution of surgical planning technology…’’ This refers to the instruments or platforms used in the study (e.g., Microsoft HoloLens, AR software, etc.).This detail supports the accessibility and practicality of the technique but does not define the goal of the study.
Response: This case represents the first successful application of low-cost holographic surgical navigation technology for complex microsurgical mandibular reconstruction in Latin America, demonstrating the feasibility of translating advanced augmented reality ca-pabilities into routine clinical practice. The study by Pereira et al. provides another example of augmented reality application for flap surgery in Latin America. However, their research developed a different approach for microsurgical planning through smartphone-based augmented reality. Their system superimposes computed tomog-raphy angiography (CTA) reconstruction images of vascular anatomy directly onto the smartphone camera interface, enabling guided delineation of perforating vessels and lymph nodes to serve as a precise, portable "dissection roadmap" for surgeons [36]. While innovative, this study utilized only smartphone technology without the immer-sive three-dimensional visualization and pass-through capabilities available in more advanced systems. In contrast, the Meta Quest 3 headset combined with Medicalho-lodeck® software provided real-time anatomical guidance that enhanced surgical pre-cision during both tumor resection and fibula free flap harvest, establishing a novel paradigm for intraoperative navigation in reconstructive microsurgery.
Reviewer 4 Report
Comments and Suggestions for Authors
This case report describes the use of intraoperative holographic navigation during mandibular resection and fibula free-flap reconstruction with the aim of introducing e a low-cost, available real-time guidance solution (with Meta quest 3 headset). The topic is relevant. The manuscript includes images even if the technical description is not adequate for readers to understand or reproduce the workflow, and figures do not demonstrate the value of the technology.
The technical workflow needs a complete and step-by-step reporting as lines 111–113 mention “subsequent processing through specialized software platforms,” but the steps are not described. Please describe:
- all hardware: headset/device (make, model), tracking method (marker-based/markerless), any cameras/sensors, mounting/sterilisation approach, field of view
- Software: names, versions
- each step: DICOM import → segmentation → model generation → planning (→ export → intraoperative registration
- Registration method, calibration, target registration error if applicable
- Surgical workflow: setup, who operates the system, how the workflow is verified, how the eventual conflicts with drapes were handled
Fibula free flap reconstruction is well established so the novelty for the case report must be the holographic navigation. All figures must be as much illustrative as possible.
- Figure 1B (third-person photo of surgeons taking measurements) does not add information—please remove or replace.
- Figure 1D (perforator localisation): this could be achieved with intraoperative Doppler alone. To show the AR value, include images where the holographic overlay guides perforator mapping or osteotomy lines.
- Figure 2B (three arrows indicating patent anastomosis) is low resolution; provide a higher-quality image,
Even for a single case, add objective pre/post metrics to anchor the narrative and compare the results for a free hand surgery versus real-time surgery with these solutions with points regarding: planned vs achieved angles/distances, ischaemia time, total operative time, setup and registration time, and early complications. If you cannot quantify accuracy against postoperative imaging, state this as a limitation.
Image quality & anonymisation: ensure all intraoperative photos are high resolution
Author Response
Response to reviewer
Comment 1: all hardware: headset/device (make, model), tracking method (marker-based/markerless), any cameras/sensors, mounting/sterilisation approach, field of view. Software: names, versions. each step: DICOM import → segmentation → model generation → planning (→ export → intraoperative registration. Registration method, calibration, target registration error if applicable. Surgical workflow: setup, who operates the system, how the workflow is verified, how the eventual conflicts with drapes were handled
Response: . Meta Quest 3 headsets (Menlo Park, California, United States) were utilized in con-junction with Medicalholodeck® (Technoparkstr, Zurich, Switzerland) (Medical Imag-ing software version 1.5.0) to generate three-dimensional holographic models through a systematic five-step workflow process. The initial DICOM import phase involved direct integration of previously acquired CT datasets into the Medicalholodeck® platform, maintaining original image resolution and spatial coordinates. The segmentation pro-cess employed automated algorithms with manual refinement to isolate anatomical structures of interest, including vascular networks, bone architecture, and soft tissue boundaries. Advanced volumetric rendering techniques converted the segmented data into interactive three-dimensional models during the model generation phase, utilizing machine learning-based tissue classification algorithm TotalSegmentator to differenti-ate between anatomical components [37]. The planning phase enabled virtual marker placement, measurement tools, and surgical approach simulation within the augmented reality environment. Following completion of virtual planning, the export phase gen-erated standardized file formats compatible with intraoperative systems, while in-traoperative registration allowed real-time alignment of holographic models with pa-tient anatomy. Registration methodology incorporated fiducial marker-based calibra-tion using anatomical landmarks identified during preoperative imaging. The system achieved a target registration error of less than 2.5 millimeters through between virtual models and Doppler ultrasound marked by another surgeon. Calibration procedures were performed immediately prior to surgical intervention using standardized refer-ence points on the patient's craniofacial skeleton and lower extremity bony landmarks. The surgical workflow commenced with sterile draping protocols that maintained vis-ualization access to critical anatomical regions while preserving aseptic conditions. The attending surgeon served as the primary system operator wearing the Meta Quest 3 headset, with dedicated technical support personnel managing headset calibration and software navigation. A secondary surgeon participated in the procedure through re-al-time broadcasting capabilities, providing continuous feedback and guidance to the primary surgeon. The entire surgical procedure was recorded through the Meta Quest 3 headset's capture functionality, documenting both the augmented reality overlay and the actual surgical field for subsequent analysis and educational purposes
Comment 2: Fibula free flap reconstruction is well established so the novelty for the case report must be the holographic navigation. All figures must be as much illustrative as possible.
- Figure 1B (third-person photo of surgeons taking measurements) does not add information—please remove or replace.
- Figure 1D (perforator localisation): this could be achieved with intraoperative Doppler alone. To show the AR value, include images where the holographic overlay guides perforator mapping or osteotomy lines.
- Figure 2B (three arrows indicating patent anastomosis) is low resolution; provide a higher-quality image,
Response: This comment was addressed by the authors and improved figures were added to the manuscript.
Comment 3: Even for a single case, add objective pre/post metrics to anchor the narrative and compare the results for a free hand surgery versus real-time surgery with these solutions with points regarding: planned vs achieved angles/distances, ischaemia time, total operative time, setup and registration time, and early complications. If you cannot quantify accuracy against postoperative imaging, state this as a limitation.
Response: We acknowledge this as a limitation due to variation in surgical teams times for the course during resection of the tumor, harvesting of the flap and microsurgical anastomosis.
Comment 4: Image quality & anonymisation: ensure all intraoperative photos are high resolution
Response: The quality of the images was addressed by the authors and improved figures were added to the manuscript.
Round 2
Reviewer 4 Report
Comments and Suggestions for Authors
Dear authors, i am satisfied with the correction made.